# Utility of data from the Danish National School Test Program for health research purposes: Content and associations with sociodemographic factors and higher education

**Anders H. Hjulmand**[1,2,3,4]*, **Betina B. Trabjerg**[1,2], **Julie W. Dreier**[1,2], **Jakob Christensen**[3,4]

1 National Centre for Register-Based Research, Aarhus BSS, Aarhus University, Aarhus, Denmark,
2 Centre for Integrated Register-based Research (CIRRAU), Aarhus University, Aarhus, Denmark,
3 Department of Neurology, Aarhus University Hospital, Aarhus, Denmark, 4 Department of Clinical Medicine, Aarhus University, Aarhus, Denmark

* ahh.ncrr@au.dk

**Data Availability Statement:** Data were based on Danish national registers, and individual-level data cannot be shared due to national regulations. The

## Abstract

The Danish National School Test Program is a set of nationwide tests performed annually since 2010 in all public schools in Denmark. To assess the utility of this data resource for health research purposes, we examined the association of school test performance with demographic and socioeconomic characteristics as well as correlations with ninth-grade exams and higher educational attainment. This nationwide descriptive register-based study includes children born between 1994 and 2010 who lived in Denmark at the age of six years. Norm-based test scores (range 1–100, higher scores indicate better performance) in reading (Danish) and mathematics from the Danish National School Test Program were obtained for children aged 6–16 attending public schools in Denmark from 2010 to 2019. Population registers were used to identify relevant demographic and socioeconomic variables. Mean test scores by demographic and socioeconomic variables were estimated using linear regression models. Among the full Danish population of 1,137,290 children (51.3% male), 960,450 (84.5%) children attended public schools. There were 885,360 children who completed one or more tests in reading or mathematics (test participation was 77.8% for the entire population, and 92.1% for children in public schools). Mean test scores varied by demographic and socioeconomic characteristics, most notably with education and labour market affiliation of parents. For every 1-point decrease in the test scores, there was a 0.95% (95% CI: 0.93%; 0.97%) lower probability of scoring B or higher in the ninth-grade exam and a 1.03% (95% CI: 1.00%; 1.05%) lower probability of completing high school within five years after graduating from lower secondary school. In this study of schoolchildren in Denmark, demographic and socioeconomic characteristics were associated with test scores from the Danish National School Test Program. Performance in school tests correlated closely with later educational attainment, suggesting that

data is governed by the Danish Government (fo@ft.
dk), Statistics Denmark (https://www.dst.dk/en/
TilSalg/Forskningsservice/kontakt-
forskningsservice) and the Ministry of Children and
Education (https://eng.uvm.dk/service/contact).
Original data are available upon request to the
relevant authorities. Additional results are provided
in the supplementary material.

**Funding:** This work was funded by the Novo
Nordisk Foundation (https://novonordiskfonden.dk/
) (NNF16OC0019126 and NNF22OC0075033), the
Central Denmark Region (https://www.rm.dk/om-
os/English-Deutsch/), and The Danish Epilepsy
Association (https://www.epilepsiforeningen.dk/in-
english/). The funding was received by J.C. The
funders had no role in the design and conduct of
the study; collection, management, analysis, and
interpretation of the data; preparation, review, or
approval of the manuscript; and decision to submit
the manuscript for publication.

**Competing interests:** I have read the journal's
policy and the authors of this manuscript have the
following competing interests: J.C. has received
honoraria from serving on the scientific advisory
board of UCB Nordic and Eisai AB, received
honoraria for giving lectures from UCB Nordic and
Eisai AB, and received funding for a trip from UCB
Nordic. This does not alter our adherence to PLOS
ONE policies on sharing data and materials.

these early measures of school performance are good markers of subsequent academic potential.

## Introduction

Education and academic achievement are fundamental for human resource development and the progress of society [1]. Parental socioeconomic status and educational level have been associated with academic performance in children such as school test scores [2]. The Danish National School Test Program consists of a set of nationwide tests, performed annually since 2010 by approximately 300,000 school-aged children per year attending public schools in Denmark (about 80% of children in grades 1 to 9 attended public schools in 2022) [3,4]. By linking individual, nationwide information from these children, it is possible to study the association of demographic and socioeconomic characteristics in these children and their parents with school tests and later academic achievement.

A previous study on the Danish National School Test Program found that test scores were associated with socioeconomic status and later educational achievement [5]. However, this study only included test scores from 2010 to 2013 and did not examine non-participation in the tests. Our study provides novel, descriptive evidence of the association between demographic and socioeconomic characteristics and performance in the national tests from 2010 to 2019 as well as with later educational attainment, and insights into the participation and non-participation patterns.

The Danish National School Test Program have previously been used in health research to study the consequences of childhood conditions and diseases [6–8], prenatal exposures [9–12], and in relation to parental disease [13]. Our study aims to provide a description of the Danish National School Test Program, that could be useful for health researchers that want to utilize these tests to assess standardized school performance. We compile a data resource profile of The Danish National School Test Program, by examining the school performance of 885,360 children who participated in the tests between 2010 and 2019 as well as 251,930 children with non-participation in the tests. We then assess the association between school test results and demographic and socioeconomic factors. Finally, we examined the correlation of test results across grade levels and within and between subjects and assessed the association between test scores and ninth-grade exams, and later educational attainment.

## Material and methods

### Study design, setting, and population

In this nationwide register-based study, we used the Danish Civil Registration System [14] to identify all children born between 1994 and 2010 who were living in Denmark at the age of six (i.e. year of school entry). In Denmark, all citizens have a unique personal identification number, which we used to link all children to individual-level information from other nationwide registers. We followed a total of 1,137,290 children, of whom 885,360 children had completed at least one test in reading (Danish) or mathematics from the Danish National School Test Program between 2010 and 2019 (Fig 1). All data was analyzed at Statistics Denmark and was first accessed on April 10, 2023. The authors did not have access to information that could identify individual participants during or after data collection. According to Danish law, the analysis of the data used in this study do not require informed consent from the participants

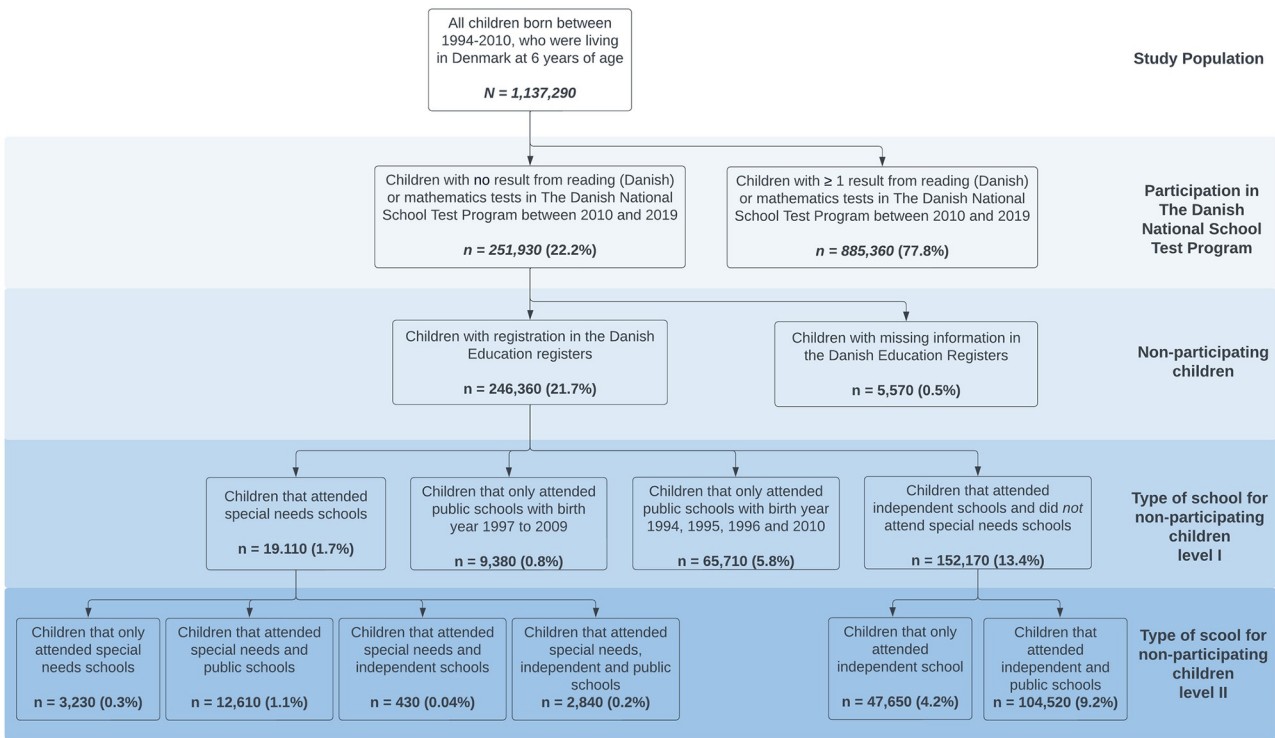

**Fig 1. Flowchart of the study population.**

or approval from an ethical review board. The study was approved by the Danish Data Protection Agency.

## Variables and data sources

**The Danish National School Test Program.** According to Danish law, all Danish children must attend compulsory education from the age of six. The Danish primary and lower secondary school comprise 10 school years, including a pre-school year (grade 0), grades 1 to 9, and an optional 10th grade. After completing primary and lower secondary school, pupils can attend upper secondary school (high school), which lasts 2 to 3 years and grants access to higher educational institutions such as universities. Attendance in Danish public schools is free of charge from preschool to university [15]. About 80% of Danish children in grades 1 to 9 attended public schools in 2022 [3]. The Danish National School Test Program was implemented in 2010 as a mandatory individual evaluation tool for children attending public schools. The test program consists of 10 compulsory tests in grades 2 to 8, covering reading (Danish), mathematics, English, and physics/chemistry. This study included tests in reading (Danish) and mathematics only. Reading ability is assessed in grades 2, 4, 6, and 8, covering 3 profile areas of language comprehension, decoding, and reading comprehension. Mathematics ability is assessed in grades 3, 6, and 8, also covering 3 profile areas of numbers and algebra, geometry, and statistics and probability. Mathematics in grade 8 was introduced in 2018. The tests are conducted online and are adaptive, meaning that the difficulty of the items adjusts to the student's performance level during the test [16]. The tests are initially scored using a Rasch model [17] that ranks individuals according to their skill level and items according to their difficulty. Test scores are then converted by the Ministry of

Education into a score between 1 and 100, where higher scores reflect better performance. The score is norm-based and reflects the student's performance as a percentile according to the nationwide score distribution on the same test in the 2010 pilot test consisting of 15,000–22,000 students [5]. The norm-based grading enables monitoring performance over time on a general level (e.g. compare the reading performance of all sixth grade students in 2014 and 2016) and on an individual level (e.g. compare the mathematics performance of a single student in third and sixth grade). In this study, we included all compulsory tests in reading (Danish) and mathematics from the years 2010 to and including 2019. We included a variable indicating whether the child was taking the test late for their age [if the test date > date of birth + 6 years (expected age at school entry) + grade + 1 year (since tests are completed at the end of a school year)].

**Demographic characteristics.** To examine variation in school performance according to demographic characteristics, we included information from the Danish Civil Registration System [14] on the child's age, sex, birth order, country of birth, as well as maternal and paternal country of birth and age at child's birth. Country of birth was divided into Danish, western, and non-western. Western origin included countries in the European Economic Area (EEA), Switzerland, Andorra, San Marino, Vatican City, United States, Canada, Australia, and New Zealand [18].

**Socioeconomic characteristics.** Maternal and paternal socioeconomic variables were ascertained in the year of school entry (i.e. the year of the child's 6th birthday), and included the highest level of completed education (primary and lower secondary, high school or vocational, short- or medium cycle higher education, and long-cycle higher education or PhD) [19], labour market affiliation (social support, retired or other, enrolled in education, self-employed, and employed) [20], equivalized disposable household income [21], (adjusted for inflation according to consumer price index in Denmark in 2016 [22], and divided into quintiles with missing values in a separate category), and maternal marital status (single, divorced or widowed, and married or cohabiting) [23].

**Ninth-grade examinations and higher education.** Information on ninth-grade exam results in reading (Danish) and mathematics was obtained up until 2019 [19] and converted into European Credit Transfer System grades (ECTS), which consists of six levels (A, B, C, D, E, F) [24]. The ninth-grade language (Danish) exam consists of four subtests in reading, spelling, writing, and verbal abilities. The mathematics exam consists of a subtest without aids and a subtest with aids. A weighted average was used to combine subtest grades from the ninth-grade exam into one grade for language and mathematics, respectively. In the case of a missing subtest, the remaining weights were adjusted proportionally to their respective sizes. Information on higher educational attainment was obtained [19] for the sub-set of children with an 8th grade reading (Danish) test in 2012 and children with a 6th grade mathematics test in 2010. These sub cohorts were selected to ensure that all children had 5 years to complete high school after graduating from lower secondary school.

**The type of school for non-participating children.** We obtained information on whether children that did not participate in the Danish National School Test Program attended an independent school or a special needs school from the Danish Education Registers [19]. Independent schools offer an alternative to public schools and have a wide range of educational styles. Special needs schools, on the other hand, cater to children with learning difficulties who require special support [25]. Children attending independent and special needs schools may participate in voluntary tests, but this study did not include voluntary school tests.

## Statistical analyses

We estimated mean test scores in each subject according to grade level, profile area, and year of testing. We then estimated mean test scores in reading (Danish) and mathematics (any grade) along with corresponding 95% confidence intervals (Cis) according to demographic and socioeconomic characteristics using linear regression models. Generalized estimating equations (GGEs) were used to obtain robust standard errors, accounting for the lack of independence of test results within each child. Separate models were fitted for reading and mathematics tests. A separate missing data category was created for each variable.

We then estimated the correlation of test results in each subject and grade with previous test results using Pearson's correlation coefficient.

We then calculated the mean change in the percentage of children who scored a B or higher in the ninth-grade exam or graduated from high school according to a 1-point decrease in test scores using linear regression models. As a supplementary analysis, we counted the number of children with test results below average (1–35) vs average and above average (36–100) according to ninth-grade exam results and highest educational level [5].

## Results

### Population

Among 1,137,290 children (51.3% male) born between 1994 and 2010 who lived in Denmark at six years of age, 885,360 (77.8%) completed one or more tests in reading (Danish) or mathematics from the Danish National School Test Program. A total of 251,930 (22.2%) children did not complete any tests, and in 246,360 (21.7%) children we were able to obtain information from the Danish Education Register regarding their school attendance, and in 5,570 (0.5%) we were not. Likely reasons for children not participating in the tests were because they attended an independent school (n = 152,170; 13.4%), for unknown reasons (i.e., attending public school but were missing a result) (n = 75,090; 6.6%), or attended a special needs school (n = 19,110; 1.7%) (Fig 1).

The proportion of children that did not complete any school tests according to birth year was highest in 1994 (89.9%), 1995 (35.9%), 1996 (25.0%), and 2010 (27.9%) as these children were eligible to participate in only one school test if they were of expected school age. For the remaining children born between 1997 to 2009, who were eligible to participate in two or more school tests, the proportion who had missing school tests was on average 6.8% (Fig 2). Furthermore, in the subset of children born between 1997 to 2009 who attended a public school (n = 743,010), the proportion with missing tests was even lower (n = 9,380; 1.3%). Compared to participating children, non-participating children were generally more likely to be of non-Danish origin, have parents of non-Danish origin, have parents with higher educational levels and that were self-employed. In addition, non-participating children had a larger proportion of missing information on demographic and socioeconomic characteristics (Tables 2 and 3). Nonetheless, the percentage of children with missing information did not exceed 4.3% for any of the variables.

### Descriptive data

Among 3,175,080 reading and mathematics tests registered in the Danish National School Test Program between 2010 and 2019, there were 2,060,700 reading tests and 1,114,380 mathematics tests. The mean age (SD) of children at the time of testing spanned from 8.9 (0.4) years in grade 2 to 14.9 (0.4) years in grade 8. Mean test scores in each subject varied by grade, profile

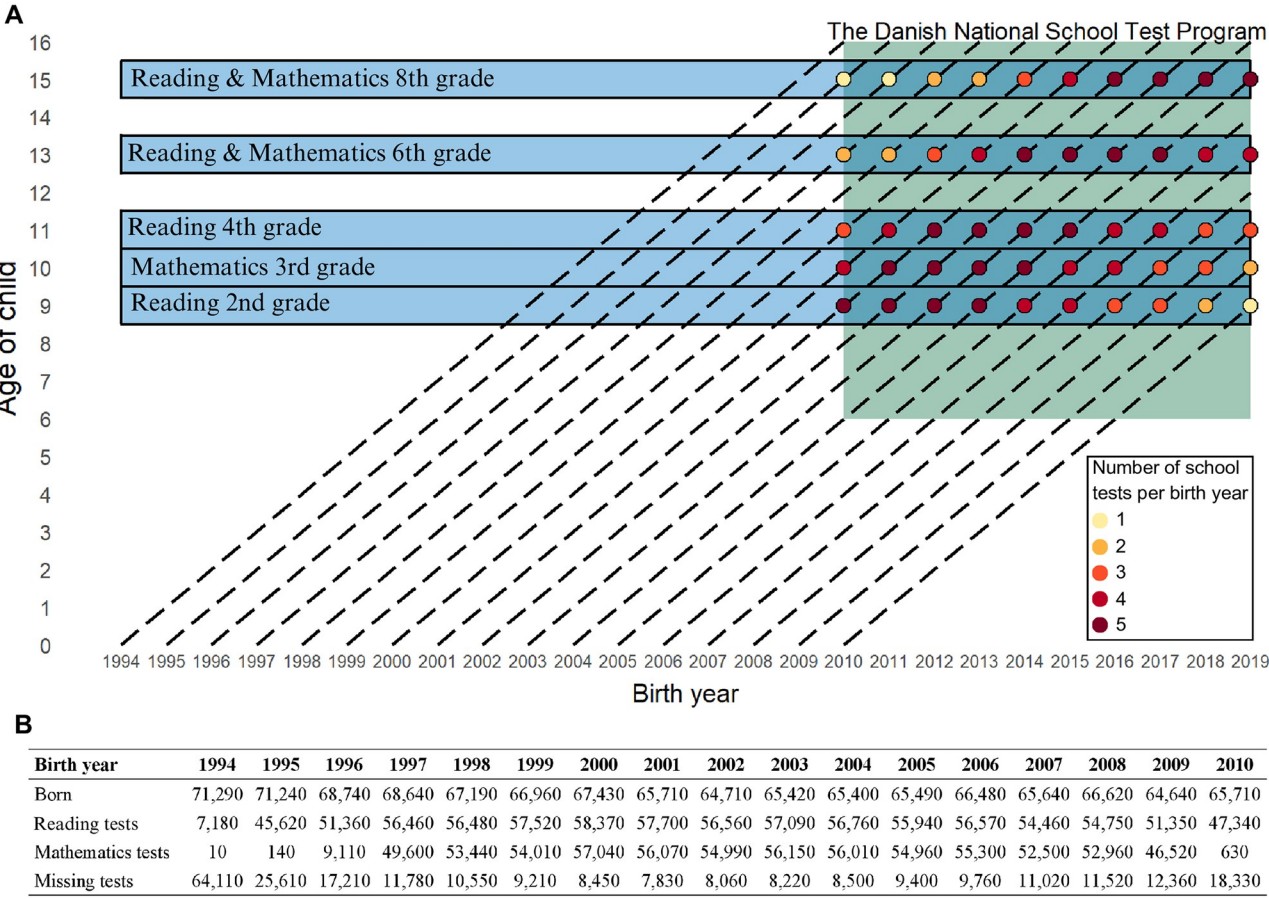

**Fig 2. Lexis diagram of birth year and test participation.** Lexis diagram of the number of tests from the Danish National School Test Program between 2010 and 2019 available for school-aged children born between 1994 and 2010 (A). Number of children that were born, that completed a reading test, that completed a mathematics test, and that did not complete a school test between 1994 and 2010 depending on birth year (B).

area, and year of testing. Mean test scores were generally higher in the late calendar years but did not increase uniformly (Table 1).

## Main results

**Child demographic and test-related characteristics.** Children born in later years, with higher birth order, of Danish or western origin, and who did not take the test at a late age, had higher mean test scores in reading and mathematics. For instance, the difference in mean test scores between firstborn and fifthborn or higher was 16 score points, and the difference between normal-aged children and children who participated in a test at a late age was 18 score points. Females had higher mean test scores in reading but not in mathematics compared to males (Table 2).

**Parental demographic and socioeconomic characteristics.** Parental higher age at the child's birth, higher levels of education, employment (employed or self-employed), enrolment in education, higher household income, civil status (married or cohabiting), and Danish origin were associated with higher mean test scores in reading and mathematics.

For instance, the difference in mean test scores in reading and mathematics between children of mothers with the lowest and the highest educational attainment was 25 to 27 score

**Table 1. Number of tests and mean test scores in reading (Danish) and mathematics tests according to grade, profile area, and year of testing.**

| Grade | Reading tests (Danish) N = 2,060,700 | | | | | | | | Mathematics tests N = 1,114,380 | | | | | |
|---|---|---|---|---|---|---|---|---|---|---|---|---|---|---|
| | 2nd | | 4th | | 6th | | 8th | | 3rd | | 6th | | 8th a | |
| Characteristics | n | Mean score | n | Mean score | n | Mean score | n | Mean score | n | Mean Score | n | Mean score | n | Mean Score |
| Overall | 528,780 | 55.2 | 527,410 | 53.7 | 521,060 | 54.4 | 483,440 | 55.9 | 529,390 | 55.4 | 519,820 | 57.1 | 95,180 | 50.4 |
| Profile area [b] | | | | | | | | | | | | | | |
| 1st | 528,780 | 55.4 | 527,410 | 53.8 | 521,060 | 55.2 | 483,440 | 52.9 | 529,390 | 57.2 | 519,820 | 56.8 | 95,180 | 49.9 |
| 2nd | 528,780 | 55.1 | 527,410 | 49.2 | 521,060 | 51.9 | 483,440 | 56.0 | 529,390 | 55.8 | 519,820 | 56.8 | 95,180 | 50.4 |
| 3rd | 528,780 | 55.1 | 527,410 | 58.2 | 521,060 | 56.1 | 483,440 | 58.8 | 529,390 | 53.2 | 519,820 | 57.5 | 95,180 | 50.9 |
| Year | | | | | | | | | | | | | | |
| 2010 | 48,220 | 51.4 | 48,640 | 51.1 | 49,390 | 51.7 | 44,640 | 50.2 | 49,780 | 51.3 | 49,520 | 51.6 | - | - |
| 2011 | 53,130 | 54.6 | 54,560 | 54.6 | 53,600 | 54.9 | 49,320 | 53.0 | 53,660 | 51.6 | 53,320 | 55.8 | - | - |
| 2012 | 54,730 | 56.3 | 53,690 | 53.2 | 53,660 | 56.2 | 50,650 | 57.2 | 52,780 | 51.8 | 53,620 | 56.0 | - | - |
| 2013 | 54,320 | 57.6 | 51,890 | 55.8 | 53,130 | 57.7 | 48,680 | 59.3 | 53,580 | 52.9 | 52,920 | 56.7 | - | - |
| 2014 | 53,630 | 57.8 | 53,850 | 56.3 | 52,420 | 58.0 | 49,350 | 59.9 | 54,340 | 53.2 | 52,280 | 57.3 | - | - |
| 2015 | 55,020 | 54.8 | 54,060 | 54.9 | 51,140 | 55.3 | 49,770 | 56.0 | 53,750 | 58.8 | 51,060 | 60.0 | - | - |
| 2016 | 53,270 | 55.3 | 52,800 | 53.4 | 52,380 | 54.1 | 48,420 | 54.9 | 54,450 | 56.9 | 52,260 | 59.9 | - | - |
| 2017 | 53,840 | 55.1 | 53,590 | 53.2 | 52,350 | 53.1 | 46,850 | 55.9 | 52,730 | 59.6 | 52,230 | 60.3 | - | - |
| 2018 | 51,570 | 55.1 | 51,980 | 52.7 | 51,180 | 51.9 | 48,140 | 56.3 | 53,240 | 58.9 | 51,030 | 56.3 | 47,830 | 50.4 |
| 2019 | 51,000 | 53.7 | 52,350 | 51.5 | 51,830 | 50.8 | 47,610 | 55.8 | 51,070 | 58.7 | 51,560 | 56.5 | 47,340 | 50.5 |
| No test result | 1,230 | - | 1,100 | - | 1,240 | - | 2,040 | - | 1,380 | - | 1,360 | - | 0 | - |
| Exempted | 170 | - | 160 | - | 180 | - | 290 | - | 140 | - | 180 | - | 0 | - |

[a] Mathematics tests in grade 8 were introduced in 2018.

[b] The profile areas of reading tests are language comprehension (1st), decoding (2nd), and reading comprehension (3rd). The profile areas of mathematics tests are numbers and algebra (1st), geometry (2nd), and statistics and probability (3rd).

points, and the difference between children of fathers that were of Danish and non-Danish origin was 3 to 10 score points (Table 3).

**Association between test results and previous test results.** There were moderate to high correlations between test results in each subject and grade, with Pearson's correlation coefficient ranging from 0.48 to 0.77. The correlation coefficient was generally higher for same-subject tests. Additionally, correlations were generally stronger for tests that were temporally closer. For example, the correlation between reading 2nd grade and 4th grade was 0.71 whereas the correlation between reading 2nd grade and 6th grade was 0.65 (Fig 3).

**Correlation between school tests, ninth-grade final exams, and higher education.** Among 827,290 children with a test result from the Danish National School Test Program between 2010 and 2018, 471,360 (57.0%) had data from their ninth-grade exam up until 2019. We found a strong association between test scores and ninth-grade exam results (Fig 4A). A decrease of 1 point in mean test scores across all tests in each child was associated with a 0.95% (95% CI: 0.93; 0.97%) lower probability of obtaining a B or higher in the ninth-grade final exam (S1 Fig). Test scores of 50,650 children with an 8th grade reading test in 2012 and 49,520 children with a 6th grade mathematics test in 2010 were strongly associated with later educational attainment (Fig 4B). A decrease of 1 point in test scores of these children was associated with a 1.03% (95% CI: 1.00%; 1.05%) lower probability of completing high school within five years after graduating lower secondary school (S2 Fig). Note that the relationship between test scores, ninth-grade exam results, and later educational attainment was not linear. We also

**Table 2. Mean test scores by demographic and test-related characteristics of school-aged children with and without a test.**

| Characteristics | Children with no school tests | | Children with one or more school tests | | | | | | | |
| --- | --- | --- | --- | --- | --- | --- | --- | --- | --- | --- |
| | | | Reading tests (Danish) | | | | Mathematics tests | | | |
| | *N* = 251,930 | | *N* = 881,510 | | | | *N* = 709,450 | | | |
| | n | (%) | n | (%) | Mean score[a] | (95% CI)[b] | n | (%) | Mean score[a] | (95% CI)[b] |
| Birth year | | | | | | | | | | |
| 1994–1997 | 118,710 | (47.1) | 160,620 | (18.2) | 52.7 | (52.7; 52.8) | 58,850 | (8.3) | 51.1 | (51.1; 51.2) |
| 1998–2001 | 36,040 | (14.3) | 230,060 | (26.1) | 55.3 | (53.3; 55.4) | 220,560 | (31.1) | 54.6 | (54.6; 54.7) |
| 2002–2005 | 34,180 | (13.6) | 226,350 | (25.7) | 55.4 | (55.3; 55.3) | 222,120 | (31.3) | 55.5 | (55.4; 55.5) |
| 2006–2010 | 63,000 | (25.0) | 264,480 | (30.0) | 54.0 | (54.0; 54.1) | 207,920 | (29.3) | 58.7 | (58.6; 58.8) |
| Sex | | | | | | | | | | |
| Male | 128,860 | (51.1) | 451,610 | (48.8) | 52.5 | (52.4; 52.5) | 363,200 | (48.8) | 56.0 | (55.9; 56.0) |
| Female | 123,070 | (48.9) | 429,910 | (51.2) | 57.2 | (57.2; 57.3) | 346,250 | (51.2) | 55.5 | (55.4; 55.5) |
| Maternal Birth order | | | | | | | | | | |
| First | 111,630 | (44.3) | 386,510 | (43.8) | 56.9 | (56.9; 57.0) | 310,390 | (43.8) | 57.6 | (57.5; 57.6) |
| Second | 90,970 | (36.1) | 327,030 | (37.1) | 54.5 | (54.5; 54.5) | 263,800 | (37.2) | 55.9 | (55.8; 55.9) |
| Third | 35,660 | (14.2) | 123,920 | (14.1) | 52.1 | (52.0; 52.2) | 99,670 | (14.0) | 53.1 | (53.0; 53.1) |
| Fourth | 9,410 | (3.7) | 30,590 | (3.5) | 47.4 | (47.2; 47.5) | 24,690 | (3.5) | 47.7 | (47.5; 47.8) |
| Fifth or higher | 3,840 | (1.5) | 12,940 | (1.5) | 41.4 | (41.2; 41.6) | 10,480 | (1.5) | 41.4 | (41.2; 41.6) |
| Missing | 410 | (0.2) | 520 | (0.1) | 42.1 | (41.1; 43.1) | 430 | (0.1) | 45.2 | (44.0; 46.4) |
| Origin[c] | | | | | | | | | | |
| Non-western | 6,550 | (2.6) | 16,280 | (1.8) | 48.3 | (48.1; 48.5) | 12,360 | (1.7) | 48.6 | (48.4; 48.9) |
| Western | 8,140 | (3.2) | 13,040 | (1.5) | 54.3 | (54.1; 54.5) | 10,460 | (1.5) | 56.4 | (56.2; 56.7) |
| Danish | 237,100 | (94.1) | 852,080 | (96.7) | 54.9 | (54.9; 54.9) | 686,560 | (96.8) | 55.8 | (55.8; 55.9) |
| Missing | 150 | (0.1) | 120 | (0.0) | 47.9 | (45.8; 50.0) | 70 | (0.0) | 46.9 | (44.0; 49.8) |
| Late age at test[d] | | | | | | | | | | |
| Yes | - | - | 35,930 | (1.1) | 36.8 | (36.6; 37.0) | 33,520 | (1.1) | 37.5 | (37.3; 37.8) |
| No | - | - | 3,165,220 | (98.9) | 55.0 | (55.0; 55.0) | 2,987,300 | (98.9) | 55.9 | (55.9; 56.0) |

[a] The score is norm-based and reflects the student's performance as a percentile of the nationwide score distribution in the same test in the 2010 pilot program. However, mean test scores varied in the subsequent years.

[b] Linear regression models with generalized estimating equations were used to obtain robust standard errors, accounting for the correlation of test scores within each child.

[c] Western origin included countries in the European Economic Area (EEA), Switzerland, Andorra, San Marino, Vatican City, United States, Canada, Australia, and New Zealand.

[d] Late age at test is assessed for each child and test.

**Abbreviations:** CI, confidence intervals.

found a strong association between test scores, ninth-grade exam results and later educational attainment in the supplemental analysis (S1 Table).

## Discussion

### Key results and interpretation

In this nationwide register-based study of more than 1,135,000 school-aged children, demographic and socioeconomic characteristics were associated with performance in the Danish National School Test Program. This suggests that it may be relevant to include these characteristics as covariates when using test scores from the Danish National School Test Program.

**Table 3. Mean test scores by demographic and socioeconomic characteristics of parents to school-aged children with and without a test.**

| Characteristics | Children with no school tests | | Children with one or more school tests | | | | | | | |
| --- | --- | --- | --- | --- | --- | --- | --- | --- | --- | --- |
| | | | Reading tests (Danish) | | | | Mathematics tests | | | |
| | N = 251,930 | | N = 881,510 | | | | N = 709,450 | | | |
| | n | (%) | n | (%) | Mean score[a] | (95% CI)[b] | n | (%) | Mean score[a] | (95% CI)[b] |
| Maternal age at birth, years | | | | | | | | | | |
| ≤ 24 | 36,880 | (14.6) | 126,940 | (14.4) | 46.6 | (46.5; 46.6) | 98,070 | (13.8) | 47.2 | (47.1; 47.2) |
| 25–34 | 170,940 | (67.9) | 612,830 | (69.5) | 55.9 | (55.8; 55.9) | 494,880 | (69.8) | 57.1 | (57.0; 57.1) |
| ≥ 35 | 43,880 | (17.4) | 141,470 | (16.0) | 57.2 | (51.7; 57.2) | 116,280 | (16.4) | 57.1 | (57.0; 57.2) |
| Missing | 240 | (0.1) | 280 | (0.0) | 39.7 | (38.8; 41.1) | 210 | (0.0) | 40.1 | (38.6; 41.6) |
| Paternal age at birth, years | | | | | | | | | | |
| ≤ 24 | 16,680 | (6.6) | 58,690 | (6.7) | 46.1 | (46.0; 46.2) | 45,240 | (6.4) | 46.2 | (46.1; 46.3) |
| 25–34 | 150,380 | (59.7) | 542,530 | (61.7) | 55.1 | (55.1; 55.2) | 436,380 | (61.5) | 56.4 | (56.4; 56.5) |
| ≥ 35 | 82,440 | (32.7) | 273,060 | (31.0) | 56.0 | (55.9; 56.0) | 223,010 | (31.4) | 56.3 | (56.3; 56.4) |
| Missing | 2,420 | (1.0) | 6,230 | (0.7) | 50.4 | (50.1; 50.7) | 4,820 | (0.7) | 48.9 | (48.5; 49.3) |
| Maternal education[c] | | | | | | | | | | |
| Primary and lower secondary | 46,330 | (18.4) | 157,080 | (17.8) | 43.5 | (43.4; 43.6) | 112,840 | (17.3) | 43.3 | (43.2; 43.4) |
| High school or vocational | 96,720 | (38.4) | 364,280 | (41.3) | 52.3 | (52.2; 52.3) | 291,630 | (41.1) | 53.2 | (53.2; 53.2) |
| Short- or medium cycle higher education | 71,570 | (28.4) | 254,890 | (28.9) | 60.8 | (60.7; 60.8) | 209,170 | (29.5) | 61.8 | (61.8; 70.2) |
| Long-cycle higher education or PhD | 28,820 | (11.4) | 89,490 | (10.2) | 68.9 | (68.8; 69.0) | 73,620 | (10.4) | 70.3 | (70.2; 70.3) |
| Missing | 8,480 | (3.4) | 15,780 | (1.8) | 45.4 | (45.2; 45.6) | 12,190 | (1.7) | 46.9 | (46.7; 47.1) |
| Paternal education[c] | | | | | | | | | | |
| Primary and lower secondary | 44,270 | (17.6) | 162,290 | (18.4) | 45.6 | (45.5; 45.7) | 128,490 | (18.1) | 45.4 | (45.3; 45.5) |
| High school or vocational | 107,780 | (42.8) | 407,850 | (46.3) | 52.9 | (52.9; 53.0) | 327,710 | (46.2) | 54.0 | (54.0; 54.1) |
| Short- or medium cycle higher education | 51,030 | (20.3) | 176,420 | (20.0) | 61.2 | (61.1; 61.2) | 114,080 | (20.3) | 62.4 | (62.4; 62.5) |
| Long-cycle higher education or PhD | 33,890 | (13.5) | 100,990 | (11.5) | 67.9 | (67.9; 68.0) | 82,420 | (11.6) | 69.3 | (69.2; 69.5) |
| Missing | 14,960 | (5.9) | 33,960 | (3.9) | 47.7 | (47.5; 47.8) | 26,750 | (3.8) | 47.3 | (47.1; 47.4) |
| Maternal labor market affiliation | | | | | | | | | | |
| Social Support | 43,310 | (17.2) | 139,190 | (15.8) | 45.1 | (45.0; 45.1) | 109,510 | (15.4) | 44.7 | (44.6; 44.7) |
| Retired or other | 12,520 | (5.0) | 25,120 | (2.8) | 49.3 | (49.1; 49.4) | 19,720 | (2.8) | 50.4 | (50.2; 50.6) |
| Enrolled in education | 8,360 | (3.3) | 25,500 | (2.9) | 56.4 | (56.2; 56.5) | 20,360 | (2.9) | 55.5 | (55.3; 55.7) |
| Self-employed | 11,210 | (4.4) | 28,920 | (3.3) | 56.8 | (56.6; 56.9) | 23,290 | (3.3) | 57.9 | (57.8; 58.1) |
| Employed | 175,580 | (69.7) | 660,620 | (74.9) | 56.8 | (56.8; 56.8) | 534,860 | (75.4) | 58.0 | (58.0; 58.0) |
| Missing | 940 | (0.4) | 2,170 | (0.2) | 47.2 | (46.7; 47.7) | 1,710 | (0.2) | 48.4 | (47.7; 49.0) |
| Paternal labor market affiliation | | | | | | | | | | |
| Social support | 23,520 | (9.3) | 78,940 | (9.0) | 43.7 | (43.7; 43.8) | 62,860 | (8.9) | 43.4 | (43.3; 43.5) |
| Retired and other | 6,170 | (2.4) | 16,710 | (1.9) | 50.5 | (50.3; 50.7) | 13,790 | (1.9) | 49.7 | (49.4; 49.9) |
| Enrolled ineducation | 2,520 | (1.0) | 6,610 | (0.7) | 57.3 | (57.0; 57.6) | 5,250 | (0.7) | 56.2 | (55.9; 56.6) |
| Self-employed | 24,520 | (9.7) | 69,930 | (7.9) | 54.3 | (54.2; 54.3) | 55,960 | (7.9) | 55.7 | (55.6; 55.8) |
| Employed | 189,250 | (75.1) | 694,570 | (78.8) | 56.2 | (56.2; 56.3) | 559,960 | (78.9) | 57.4 | (57.3; 57.4) |
| Missing | 5,950 | (2.4) | 14,770 | (1.7) | 49.5 | (49.3; 49.7) | 11,630 | (1.6) | 47.9 | (47.7; 48.2) |
| Household income, quintiles[d] | | | | | | | | | | |
| Lowest | 57,730 | (22.9) | 167,090 | (19.0) | 45.7 | (45.7; 45.8) | 129,470 | (18.2) | 45.6 | (45.5; 45.7) |
| Second | 50,880 | (20.2) | 174,460 | (19.8) | 50.6 | (50.6; 50.7) | 135,460 | (19.1) | 50.3 | (50.2; 50.4) |
| Third | 46,880 | (18.6) | 178,480 | (20.2) | 54.6 | (54.6; 54.7) | 143,480 | (20.2) | 55.1 | (55.0; 55.2) |
| Fourth | 43,440 | (17.2) | 182,040 | (20.7) | 58.6 | (58.6; 58.7) | 151,870 | (21.4) | 59.8 | (59.8; 59.9) |
| Highest | 49,490 | (19.6) | 175,980 | (20.0) | 63.4 | (63.3; 63.4) | 146,320 | (20.6) | 65.9 | (65.8; 65.9) |
| Missing | 3,510 | (1.4) | 3,580 | (0.4) | 49.1 | (48.6; 49.5) | 2,850 | (0.4) | 50.1 | (49.7; 50.6) |
| Maternal marital status | | | | | | | | | | |

*(Continued)*

**Table 3.** (Continued)

| Characteristics | Children with no school tests | | Children with one or more school tests | | | | | | | |
|---|---|---|---|---|---|---|---|---|---|---|
| | | | Reading tests (Danish) | | | | Mathematics tests | | | |
| | N = 251,930 | | N = 881,510 | | | | N = 709,450 | | | |
| | n | (%) | n | (%) | Mean score[a] | (95% CI)[b] | n | (%) | Mean score[a] | (95% CI)[b] |
| Single, divorced or widowed | 37,540 | (14.9) | 142,890 | (16.2) | 50.2 | (50.1; 50.2) | 115,520 | (16.3) | 49.0 | (49.0; 49.1) |
| Married or cohabiting | 211,520 | (84.0) | 733,260 | (83.2) | 55.7 | (55.7; 55.7) | 589,760 | (83.1) | 57.1 | (57.0; 57.1) |
| Missing | 2,890 | (1.1) | 5,360 | (0.6) | 48.7 | (48.4; 49.1) | 4,160 | (0.6) | 50.7 | (50.3; 51.1) |
| Maternal origin[e] | | | | | | | | | | |
| Non-western | 38,720 | (15.4) | 114,050 | (12.9) | 46.8 | (46.7; 46.9) | 91,570 | (12.9) | 48.1 | (48.0; 48.1) |
| Western | 5,320 | (2.1) | 7,030 | (0.8) | 48.8 | (48.6; 49.1) | 5,610 | (0.8) | 53.0 | (52.6; 53.3) |
| Danish | 206,860 | (82.1) | 758,710 | (86.1) | 56.0 | (56.0; 56.1) | 610,920 | (86.1) | 56.9 | (56.9; 57.0) |
| Missing | 1,040 | (0.4) | 1,730 | (0.2) | 45.2 | (44.7; 45.8) | 1,350 | (0.2) | 47.8 | (47.1; 48.4) |
| Paternal origin[e] | | | | | | | | | | |
| Non-western | 37,850 | (15.0) | 111,080 | (12.6) | 46.6 | (46.6; 46.7) | 88,960 | (12.5) | 47.5 | (47.4; 47.6) |
| Western | 4,520 | (1.8) | 6,140 | (0.7) | 50.3 | (50.0; 50.6) | 4,860 | (0.7) | 54.3 | (54.0; 54.7) |
| Danish | 205,080 | (81.4) | 753,670 | (85.5) | 56.1 | (56.0; 56.1) | 607,280 | (85.6) | 57.0 | (57.0; 57.1) |
| Missing | 4,480 | (1.8) | 10,620 | (1.2) | 50.2 | (50.2; 50.4) | 8,340 | (1.2) | 48.9 | (48.6; 49.1) |

[a] The score is norm-based and reflects the student's performance as a percentile of the nationwide score distribution in the same test in the 2010 pilot program. However, mean test scores varied in the subsequent years.

[b] Linear regression models with generalized estimating equations were used to obtain robust standard errors, accounting for the correlation of test scores within each child.

[c] Medium-cycle higher education includes bachelor' degrees from universities and university colleges; long-cycle higher education includes master's and PhD degrees from universities.

[d] Equivalized household income at age six of the child, adjusted for inflation to prices of 2016.

[e] Western origin included countries in the European Economic Area (EEA), Switzerland, Andorra, San Marino, Vatican City, United States, Canada, Australia, and New Zealand.

**Abbreviations:** CI, confidence intervals.

The magnitude of the difference in test performance between levels of parental socioeconomic characteristics was high in our study (up to 27 out of 100 points). Like previous studies, we found that low socioeconomic status of parents was associated with poorer school performance in the children [2,26,27]. There are numerous theoretical explanations for socioeconomic inequalities in academic achievement, including cognitive abilities and personality traits such as conscientiousness [2,28,29]. The substantial association between socioeconomic factors and academic achievement found in this study, does not address underlying factors such as prior test scores and cognitive abilities, that likely reflect an interaction between gene and environmental factors which would predict both school performance and socioeconomic status [29,30]. It was not the focus of the this mainly descriptive study to explain the observed socio-economic differences.

We found moderate to high correlations between prior and later test results, suggesting that prior test scores can serve as reliable indicators of performance in later tests in The Danish National School Test Program. The correlation was strongest for tests that were temporally closer and for tests within the same subject. Nonetheless, there were still moderate correlations between scores in mathematics and reading, which may be explained by a general interest in learning. Similar results were found in previous studies [5,29,31,32].

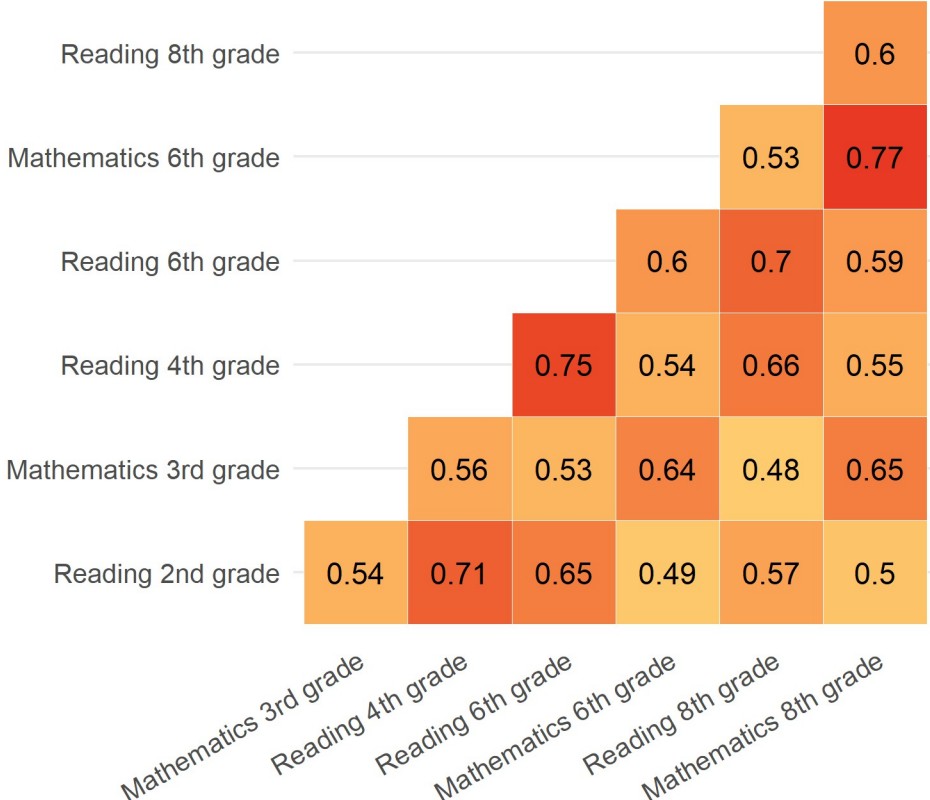

**Fig 3. Pairwise Pearson correlations between test scores in each subject and grade.**

In addition, lower test scores were strongly associated with a lower probability of obtaining a B or higher in the ninth-grade final exam and completing high school within five years after graduating lower secondary school. Our results are consistent with a previous study of the Danish National School Test Program [5]. These findings suggest that the skills assessed in the school tests, even in early grades, are closely associated with the competencies required for ninth-grade exam performance and later educational achievement.

Children who participated in a test "late for age" had an 18-point lower test score compared to children who participated at expected age. Some evidence indicates that there is a direct effect of school starting age on learning outcomes [33–35], but this finding may also suggest that "late for age" children may be more likely to have pre-existing learning difficulties, which delays school entry or result in children repeating grades.

## Strengths and limitations

This was a descriptive study of the association of socioeconomic status with school performance and there was no intention to explain the observed socio-economic differences, and the estimates cannot therefore be interpreted as "effects" that can be intervened upon. However, the present study has several strengths. First, it was based on systematically collected long-term data from the nationwide Danish registers. Second, it included individually linked information from a substantial number of children including demographic and socioeconomic variables that were highly predictive of the test scores. Third, school performance was assessed

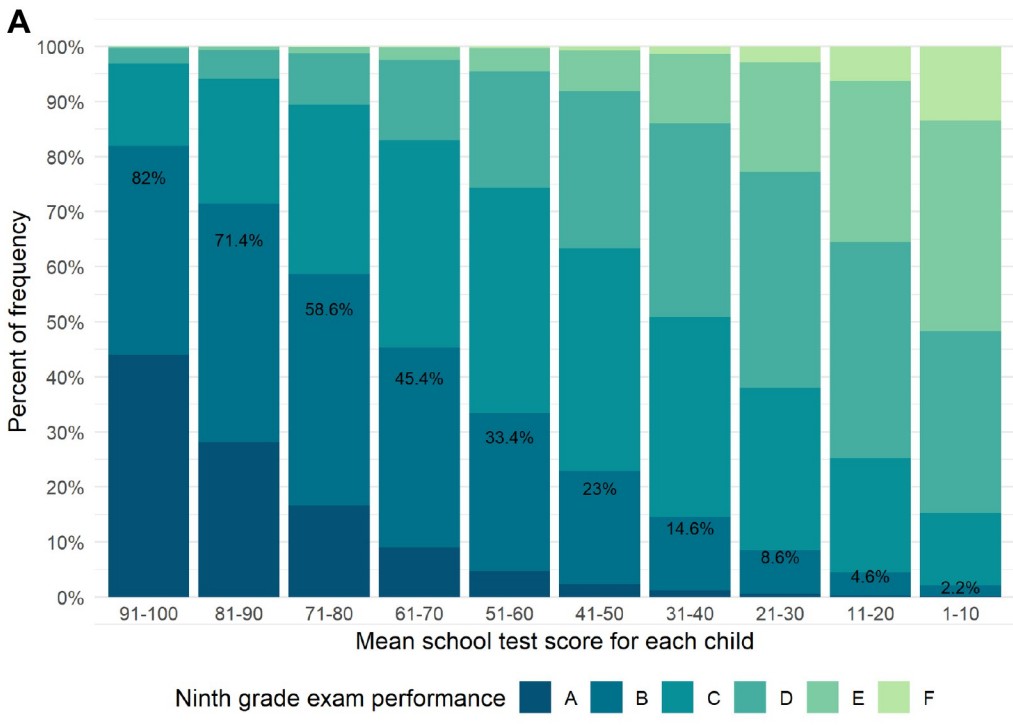

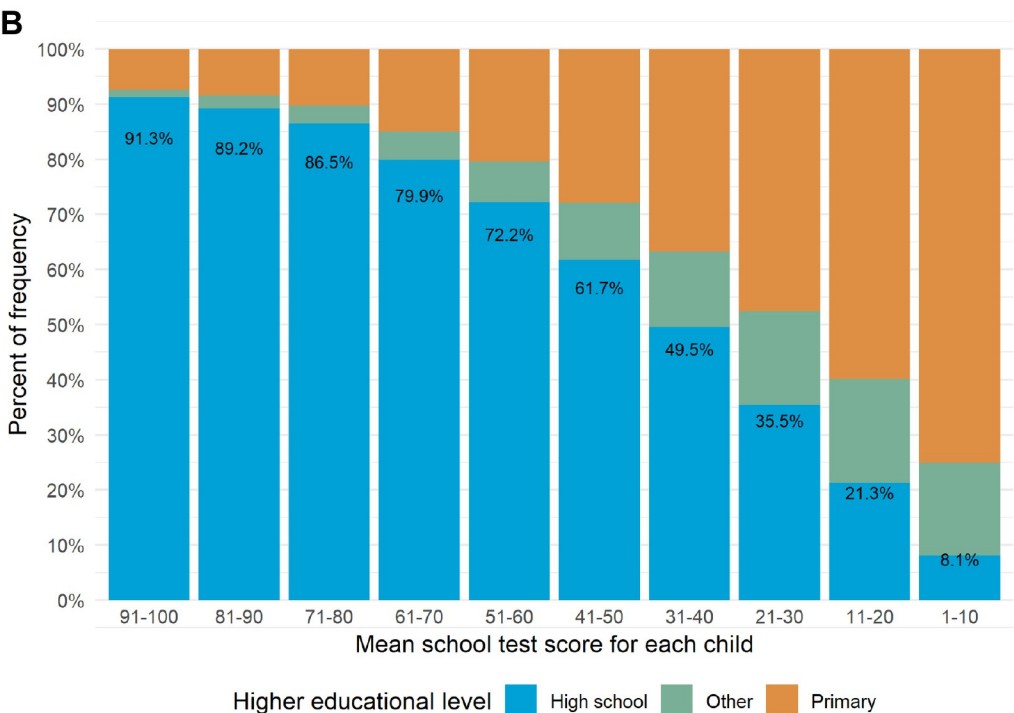

**Fig 4. European Credit Transfer System grades (ECTS) (A) and highest educational level (B) according to test scores.** Among 827,290 children with school test results from 2010 to and including 2018, 471,360 (57.0%) had information on ninth-grade exam results (A). Highest educational level within five years after graduating primary and lower secondary school was assessed for 50,650 children with an 8th grade reading test in 2012 and 49,520 children with a 6th grade mathematics test in 2010. *Other* includes children that completed an education after primary and lower secondary school but did not complete high school. *High school* includes children that completed high school and possibly other educations (B). Percentages indicate the percentage of children with grade B or higher in the ninth-grade exam (A) and the percentage of children who completed high school (B).

across many grade levels using nationwide, standardized, and automatically scored tests, thereby limiting the risk of teacher bias.

However, the study also has several limitations. The association between school performance and demographic and socioeconomic factors was examined only for children who participated in the Danish National School Test Program (77.8% of eligible children), thereby excluding children from independent or special needs schools. Non-participating children had different socioeconomic and demographic characteristics, meaning that the findings do not necessarily generalize to all children in Denmark.

Moreover, the difference in underlying skill between the 50th and the 51st percentile (i.e. in the middle of the distribution) is smaller than the difference between the 1st and 2nd percentile which is a limitation of using percentile test scores compared to Rasch test scores. However, we use percentile scores because our study is targeted towards health clinicians and researchers in epidemiology. We believe that this audience may find the percentile scores more intuitive and easier to interpret (e.g. what constitutes a large versus small score difference) than the Rasch scores. Furthermore, because of the recent completeness of the school tests (2010–2019), this study was limited by follow up time and was therefore able to only track the highest educational attainment of children within 5 years after graduating from lower secondary school. As additional data becomes available from children who participated in the Danish National School Test Program, future studies can examine associations between test performance and later socioeconomic characteristics such as educational attainment and income status.

Finally, education systems vary across countries [36], which means that the relationship between demographic and socioeconomic factors and school performance in Denmark may not necessarily generalize to other countries. In countries with a welfare state [37] and free access to public schools (including Denmark), the effect of socioeconomic background on school performance may be smaller than in other countries. Our findings may thus be limited to a Danish context.

## Conclusion

This large nationwide study of school children in Denmark describes associations of demographic and socioeconomic characteristics with performance in the Danish National School Test Program. School test performance was closely correlated with later educational attainment, suggesting that these early measures of school performance are good markers of subsequent academic potential.

## Supporting information

**S1 Fig. Percentage of children who scored B or higher in the ninth-grade final exam according to mean test scores.** Mean test scores were rounded to the nearest integer. A decrease of 1 point in the test scores was associated with a decrease of 0.95% (95% CI: 0.93%; 0.97%) in obtaining B or higher in the ninth-grade final exam. The intercept of the linear regression was 82.4% (95% CI: 81.2%; 83.5%). Although the data points did not have a linear relationship and the linear regression model predicted negative percentages for test scores below 13 points, the model predicted fairly well for test score points between 20 and 70. (PDF)

**S2 Fig. Percent of children who completed high school within five years of graduating primary and lower secondary school according to mean test scores.** Highest educational level was assessed for 50,650 children with an 8th grade reading test in 2012 and 49,520 children

with a 6th grade mathematics test in 2010. Mean test scores were rounded to the nearest integer. A decrease of 1 point in the test scores was associated with a decrease of 1.03% (95% CI: 1.00%; 1.05%) in completing high school. The intercept of the linear regression was 113.3% (95% CI: 112.0%; 114,7%). Although the data points did not have a linear relationship and the linear regression model predicted percentages above 100% for test scores above 81 points, the model predicted fairly well for test score points between 20 and 70. The following test scores were removed because there were too few children in some groups of education: 2, 89, 93, 94, 95, 96, 97, 98, 99, and 100.
(PDF)

**S1 Table. Number of children with test scores below average (1–35) vs average and above average (36–100) according to ninth-grade exam results and highest educational level.** [a] Scores are divided according to norm-referenced groups. b Information on ninth-grade final exams was obtained for children with school tests in the years 2010–2018. Grades from the Danish scale were converted into European Credit Transfer System grades (ECTS). c Highest educational level within five years after graduating primary and lower secondary school was assessed for 50,650 children with an 8th grade reading test in 2012 and 49,520 children with a 6th grade mathematics test in 2010. *Other* includes children that completed an education after primary and lower secondary school but did not complete high school. *High school or vocational* includes children that completed high school and possibly other educations.
(DOCX)

## Author Contributions

**Conceptualization:** Julie W. Dreier, Jakob Christensen.

**Formal analysis:** Anders H. Hjulmand, Betina B. Trabjerg.

**Funding acquisition:** Jakob Christensen.

**Investigation:** Anders H. Hjulmand.

**Methodology:** Anders H. Hjulmand, Betina B. Trabjerg, Julie W. Dreier, Jakob Christensen.

**Project administration:** Julie W. Dreier, Jakob Christensen.

**Resources:** Jakob Christensen.

**Software:** Anders H. Hjulmand, Betina B. Trabjerg.

**Supervision:** Julie W. Dreier, Jakob Christensen.

**Validation:** Anders H. Hjulmand, Betina B. Trabjerg, Julie W. Dreier, Jakob Christensen.

**Writing – original draft:** Anders H. Hjulmand.

**Writing – review & editing:** Betina B. Trabjerg, Julie W. Dreier, Jakob Christensen.

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
