## [Decision Letter · Decision Letter 0]

23 Oct 2023

PONE-D-23-29899The Danish National School Test Program: Correlates with Sociodemographic Factors and Prediction of Academic AchievementPLOS ONE

Dear Dr. Hjulmand,

Thank you for submitting your manuscript to PLOS ONE. After careful consideration, we feel that it has merit but does not fully meet PLOS ONE’s publication criteria as it currently stands. Therefore, we invite you to submit a revised version of the manuscript that addresses the points raised during the review process.

We look forward to receiving your revised manuscript.

Kind regards,

Daniel Ramskov, Ph.D

Academic Editor

PLOS ONE

Journal Requirements:

3. Thank you for stating the following in the Competing Interests section: "I have read the journal's policy and the authors of this manuscript have the following competing interests: J.C. has received honoraria from serving on the scientific advisory board of UCB Nordic and Eisai AB, received honoraria for giving lectures from UBC Nordic and Eisai AB, and received funding for a trip from UCB Nordic.".

Reviewers' comments:

Reviewer's Responses to Questions

**Comments to the Author**

1. Is the manuscript technically sound, and do the data support the conclusions?

Reviewer #1: Partly

Reviewer #2: Yes

Reviewer #3: Partly

2. Has the statistical analysis been performed appropriately and rigorously? 

Reviewer #1: No

Reviewer #2: Yes

Reviewer #3: Yes

3. Have the authors made all data underlying the findings in their manuscript fully available?

Reviewer #1: Yes

Reviewer #2: No

Reviewer #3: No

4. Is the manuscript presented in an intelligible fashion and written in standard English?

Reviewer #1: Yes

Reviewer #2: Yes

Reviewer #3: Yes

5. Review Comments to the Author

Reviewer #1: This study does majorly suffer from two important things:

1. A misguided description of the research literature on academic achievement, including the Danish context. There are many large-scale studies in for example Denmark, Norway and Sweden. Also, the international literature is largely missing.

2. Omitted variable bias. Considerable research shows that socioeconomic and sociodemographic factors are not strongly associated with academic achievement when cognitive ability and conscientiousness (or similar non-cognitive constructs such as academic self-concept and/or self-efficacy) are included in the same multivariate models. Hence, readers cannot have a strong confidence in your regression estimates.

If there are no such variables in your data set you must critically discuss the limitations of your results. This has not been done in the manuscript.

I suggest that you work with the data and include more variables and then re-analyze the revised models. Please do also include some of the studies that I list below:

Andersen, S.C. Gensowski, M. Ludeke, S., & John, O. (2020). A stable relationship between personality and academic performance from childhood through adolescence: An original study and replication in hundred-thousand-person samples. Journal of Personality, 88 (5), 925–939.

Boman, B. (2023). The influence of SES, cognitive and non-cognitive abilities on grades: cross-sectional and longitudinal evidence from two Swedish cohorts. European Journal of Psychology of Education, 38, 587–603.

Mammadov, S. (2022). Big Five personality traits and academic performance: a meta-analysis. Journal of Personality, 90 (2), 222–255.

Marks, G.N. (2022). Cognitive ability has powerful, widespread and robust effects on social stratification: Evidence from the 1979 and 1997 US National Longitudinal Surveys of Youth. Intelligence (September–October), 94.

Marks, G.N., & O’Connell, M. (2021). Inadequacies in the SES–achievement model: Evidence from PISA and other studies. Review of Education, 9(3).

Vazsonyi, A.T. Javakhishvili, M. Blatny, M. (2022). Does self-control outdo IQ in predicting academic performance? Journal of Youth and Adolescence, 51 (4), 499–508.

von Stumm, S., Hell, B., & Chamorro-Premuzic, T. (2011). The hungry mind: Intellectual curiosity is the third pillar of academic performance. Perspectives on Psychological Science, 6(6), 574–588.

Reviewer #2: RE: The Danish National School Test Program: Correlates with Sociodemographic Factors

and Prediction of Academic Achievement

The literature review is very light-on. There are many studies on achievement over school career (by level) and the correlations between achievement in mathematics and reading (Reynolds & Walberg, 1992, p. 318; Armor, 2003, p. 33; McNiece, Bidgood, & Soan, 2004, p. 134; Parsons, 2014, p. 36; Sullivan, Parsons, Wiggins, Heath, & Green, 2014, p. 752; Kriegbaum, Jansen, & Spinath, 2015; Ludeke et al., 2021, p. 1083; Marks, 2021). There is a huge academic literature on SES and achievement.

I am not requesting an extensive literature review, but some discussion of the results found in previous studies, which are similar to the those in presented in the reviewed paper: high and possibly increasing within domain temporal correlations and slightly weaker between-domain correlations. In their Discussion section, the authors should attempt to explain why the results are similar, or in the authors’ judgement, different from, comparable studies conducted in other countries.

There are no theoretical explanations in the paper for the patterns observed. Could the authors please discuss their results in relation to the literature on general and specific cognitive abilities or an alternative explanation they find plausible.

According to the title of the paper, it is about sociodemographic factors. My question is, do socioeconomic factors strongly impact on test scores when considering prior test scores. With prior achievement accounting for between 25 and 60% of the variance, I expect the effects of sociodemographic factors to be quite weak.

The paper needs some additional analyses with and without prior test scores vis-a-vis continuous some measures of the socioeconomic characteristics (mother’s and father’s education, and logged family income). The paper is purportedly about socioeconomic factors, but they are hardly discussed.

What are the correlations of test scores with the SES measures? Could continuous measures of father’s and mother’s education, and income be added to eFigure 1 (which should replace Figure3 in the main body). It is not clear if SES correlations are weaker in Denmark than in other countries and although the authors suggest that they are probably are weaker because of a more extensive welfare state. They suggest comparing their findings with similar studies, but they do not seem to do so.

1. I expected to find the regressions in the Supplementary material which back up the claims in the main body of the paper. They don’t seem to be there.

2. Why a B or above cutoff rather than simply Grade 9 test score which is statistically simpler and more interpretable. Why discard all the information in a normally distributed variable and arbitrarily choose a cut-off of B or higher?

3. I find the correlations in eFigure 1 more meaningful to the estimates in Figure 3.

4. The estimates presented in eTable 1 are not how estimates are usually presented. For most readers, much more interpretable estimates would be changes in test scores in metric or standard deviation units for a one-standard deviation increase prior test score. This was done for GPA in the previous paper.

5. There’s no need for confidence intervals in eTable 1. The estimates and two confidence limits are exactly the same. They just clutter the table.

6. The three cited studies (25-27) are not very relevant to the paper. I found the choice of papers a bit odd; all 3 are highly econometric.

7. Relatedly, why is the previous study of Danish National Test Program mentioned in the Discussion? The Introduction should provide a summary of the results and limitations of the previous study. The reference should provide a link to the study (https://childresearch.au.dk/fileadmin/childresearch/dokumenter/2018_1_2.pdf ).

8. The paper would be more interesting if university entrance or tertiary course status were included in the measure of educational attainment, not just completing high school.

Armor, D. J. (2003). Maximizing intelligence. New Brunswick and London: Transaction Publishers.

Kriegbaum, K., Jansen, M., & Spinath, B. (2015). Motivation: A predictor of PISA's mathematical competence beyond intelligence and prior test achievement. Learning and Individual Differences, 43, 140-148. doi:https://doi.org/10.1016/j.lindif.2015.08.026.

Ludeke, S. G., Gensowski, M., Junge, S. Y., Kirkpatrick, R. M., John, O. P., & Andersen, S. C. (2021). Does parental education influence child educational outcomes? A developmental analysis in a full-population sample and adoptee design. Journal of Personality and Social Psychology, 120(4), 1074-1090. doi:https://doi.org/10.1037/pspp0000314.

Marks, G. N. (2021). Should value-added school effects models include student- and school-level covariates? Evidence from Australian population assessment data. British Educational Research Journal, 47(1), 181–204. doi:https://doi.org/10.1002/berj.3684.

McNiece, R., Bidgood, P., & Soan, P. (2004). An investigation into using national longitudinal studies to examine trends in educational attainment and development. Educational Research, 46(2), 119-136. doi:https://doi.org/10.1080/0013188042000222412.

Parsons, S. (2014). Childhood cognition in the 1970 British cohort study. Retrieved from London: https://cls.ucl.ac.uk/wp-content/uploads/2017/07/BCS70-Childhood-cognition-in-the-1970-British-Cohort-Study-Nov-2014-final.pdf

Reynolds, A. J., & Walberg, H. J. (1992). A process model of mathematics achievement and attitude. Journal for Research in Mathematics Education, 23(4), 306-328. Retrieved from http://www.jstor.org/stable/749308.

Sullivan, A., Parsons, S., Wiggins, R., Heath, A., & Green, F. (2014). Social origins, school type and higher education destinations. Oxford Review of Education, 40(6), 739-763. doi:https://doi.org/10.1080/03054985.2014.979015.

Reviewer #3: The manuscript presents analyses of the associations between test scores from The Danish National School Test Program, socio-demographic characteristics of the students and subsequent attainment in the educational system.

I have a few comments:

1. I think it would be appropriate to mention in the introduction that the results presented in this manuscript resemble those presented by Beuchert and Nandrup (reference 16). (Please note, I think there is a typo in ref. 16 (“Rochester”).)

2. It may also be appropriate to refer to the Ludeke et al. (2021) study (see reference below).

3. The study uses the so-called percentile scores from the national tests. I think it would be appropriate to use the underlying Rasch test scores instead. Since student skills are approximately normal distributed, the difference in underlying skill between 50th and 51st percentile (i.e. in the middle of the distribution) is smaller than the difference between 1st and 2nd percentile. To ease interpretability, the Rasch test scores may be standardized with mean 0 and standard deviation 1, which is common in much of the literature using the national test scores.

4. I think it would be more accurate to refer to the test in Danish “reading (Danish)” rather than “language (Danish)” since the tests only assess reading and not verbal language skills.

5. The Danish Test program was introduced by law earlier than 2010 – but in effect from 2010 (see formulation p. 5)

6. Page 11 says that test scores were generally higher in the late years, which leads the authors to speculate that this may be due to “teach to the test”. I do not think this is in consistent with results presented in Table 1 which show that test scores increased from 2010 to 2014 (reading) or 2015 (math) and then declined until 2019.

Ludeke SG, Gensowski M, Junge SY, Kirkpatrick RM, John OP, Andersen SC. Does parental education influence child educational outcomes? A developmental analysis in a full-population sample and adoptee design. J Pers Soc Psychol. 2021 Apr;120(4):1074-1090. doi: 10.1037/pspp0000314. Epub 2020 Jun 15. PMID: 32538645.

6. PLOS authors have the option to publish the peer review history of their article (what does this mean?). If published, this will include your full peer review and any attached files.

Reviewer #1: No

Reviewer #2: **Yes: **Gary N Marls

Reviewer #3: No

---

## [Author Response · Author response to Decision Letter 0]

23 Jan 2024

Response to reviewers is addressed in the file "Response to Reviewers".

---

## [Decision Letter · Decision Letter 1]

27 Feb 2024

PONE-D-23-29899R1Utility of data from the Danish National School Test Program for health research purposes: Content and associations with sociodemographic factors and higher educationPLOS ONE

Dear Dr. Hjulmand,

Thank you for submitting your manuscript to PLOS ONE. After careful consideration, we feel that it has merit but minor issues still exist. Therefore, we invite you to submit a revised version of the manuscript that addresses the points raised during the review process.

We look forward to receiving your revised manuscript.

Kind regards,

Daniel Ramskov, Ph.D

Academic Editor

PLOS ONE

Journal Requirements:

**Additional Editor Comments:**

**ACADEMIC EDITOR.**

My decision is a minor revision. Please consider the last comments carefully and respond to them as you feel relevant. All peer-reviewers found the changes you made to the original manuscript very appropriate and comprehensive. I don´t think a second round of peer review will be necessary as long as the few remaining comments are addressed. 

Reviewers' comments:

Reviewer's Responses to Questions

**Comments to the Author**

1. If the authors have adequately addressed your comments raised in a previous round of review and you feel that this manuscript is now acceptable for publication, you may indicate that here to bypass the “Comments to the Author” section, enter your conflict of interest statement in the “Confidential to Editor” section, and submit your "Accept" recommendation.

Reviewer #1: All comments have been addressed

Reviewer #2: All comments have been addressed

Reviewer #3: (No Response)

2. Is the manuscript technically sound, and do the data support the conclusions?

Reviewer #1: Yes

Reviewer #2: Yes

Reviewer #3: Yes

3. Has the statistical analysis been performed appropriately and rigorously? 

Reviewer #1: Yes

Reviewer #2: Yes

Reviewer #3: Yes

4. Have the authors made all data underlying the findings in their manuscript fully available?

Reviewer #1: Yes

Reviewer #2: Yes

Reviewer #3: No

5. Is the manuscript presented in an intelligible fashion and written in standard English?

Reviewer #1: Yes

Reviewer #2: Yes

Reviewer #3: Yes

6. Review Comments to the Author

Reviewer #1: (No Response)

Reviewer #2: My concerns and comments have been addressed. The study's conclusions are convincing, given the comprehensive data source and the authors' analyses. The paper should be published in PLos One.

Reviewer #3: Review of PONE-D-23-29899_R1 ”Utility of data from the Danish National School Test Program”

The authors have responded to my comments on the first version of the manuscript. I have only a few comments for the revised manuscript.

1. The abstract says that 77.8% of eligible children complete one or more tests. I don’t think it is correct to say that children in independent schools and special needs schools are eligible for the test. I suggest that it says 77.8% of the full population, or – even better – how state both how many of the eligible children (in public schools) complete, and how many of the population of all children compete. This will indicate to researchers, that the test program is of very high quality in terms of response rate (but not necessarily generalizable to children in independent schools). It would also help to highlight one of the contributions of this study relative to Beuchert & Nandrup (ref. no. 5), namely the analysis of non-respondents.

2. The same comments is relevant for the last paragraph in the introduction (p. 4)

3. First line on p. 8 says that participation is voluntary for independent and special needs schools. I may be wrong, but I do not think independent schools are allowed to use the tests at all. Please double check.

4. The first paragraph in the “Population” section (p. 8-9) provides the number of non-completion (22.2%) and the reasons. However it fails to mention the 0.5% with missing information in Danish registers. It is clear from Fig. 1, but confused when reading the text.

5. The discussion mentions the “late for age” result. I agree that these children may have learning difficulties. However, there is also substantial amount of research showing that school starting age itself has a causal effect on learning (see e.g. various papers by Hans Henrik Sievertsen, Helena Skyt Nielsen and Rasmus Landersø.)

6. I think the authors should delete that last paragraph on p. 17, which speculate about teaching to the test for the following reasons: 1) it is only speculative, there is no data to support it. Many other explanations could be relevant, e.g. that teachers or parents have become better at learning children to read 2) as mentioned in my first review, the results do not increase uniformly, which speaks against the teach to test-explanation (which the manuscript now acknowledges, but this makes it unclear why to mention the explanation in the first place) 3) the last sentence suggesting that studies should adjust for calendar-year does not make sense in relation to the rest of the study because it indicates that the test scores are not valid measures of reading skills. If students improved their reading skills (and not just became better at taking the test), adjusting for calendar-year would be a mistake.

7. The limitations section now mentions the difference between the Rasch score and the percentile score (p. 18). I suggest that the manuscript presents an explanation for why the study chose to use the percentile score. The authors response to my comment about this, could be a fine explanation to include in the manuscript.

7. PLOS authors have the option to publish the peer review history of their article (what does this mean?). If published, this will include your full peer review and any attached files.

Reviewer #1: No

Reviewer #2: **Yes: **Gary N. Marks

Reviewer #3: No

---

## [Author Response · Author response to Decision Letter 1]

28 Mar 2024

Response to reviewers is addressed in the file "Response to Reviewers".

---

## [Editor Report · Decision Letter 2]

5 Apr 2024

Utility of data from the Danish National School Test Program for health research purposes: Content and associations with sociodemographic factors and higher education

PONE-D-23-29899R2

Dear Dr.HJulmand

We’re pleased to inform you that your manuscript has been judged scientifically suitable for publication and will be formally accepted for publication once it meets all outstanding technical requirements.

Kind regards,

Daniel Ramskov, Ph.D

Academic Editor

PLOS ONE